# Transferable Diffusion-based Unrestricted Adversarial Attack on Pre-trained Vision-Language Models

## ABSTRACT

Pre-trained Vision-Language Models (VLMs) have shown great ability in various Vision-Language tasks. However, these VLMs exhibit inherent vulnerabilities to transferable adversarial examples, which could potentially undermine their performance and reliability in real-world applications. Cross-modal interactions have been demonstrated to be the key point to boosting adversarial transferability, but the utilization of them is limited in existing multimodal transferable adversarial attacks. Stable Diffusion, which contains multiple cross-attention modules, possesses great potential in facilitating adversarial transferability by leveraging abundant cross-modal interactions. Therefore, We propose a Multimodal Diffusion-based Attack (MDA), which conducts adversarial attacks against VLMs using Stable Diffusion. Specifically, MDA initially generates adversarial text, which is subsequently utilized as guidance to optimize the adversarial image during the diffusion process. Besides leveraging adversarial text in calculating downstream loss to obtain gradients for optimizing image, MDA also takes it as the guiding prompt in adversarial image generation during the denoising process, which enriches the ways of cross-modal interactions, thus strengthening the adversarial transferability. Compared with pixel-based attacks, MDA introduces perturbations in the latent space rather than pixel space to manipulate high-level semantics, which is also beneficial to improving adversarial transferability. Experimental results demonstrate that the adversarial examples generated by MDA are highly transferable across different VLMs on different downstream tasks, surpassing state-of-the-art methods by a large margin.

## CCS CONCEPTS

• **Computing methodologies → Artificial intelligence**.

## KEYWORDS

Pre-trained Vision-Language Models, Diffusion-based Unrestricted Attack, Adversarial Transferability

## 1 INTRODUCTION

Recently, pre-trained Vision-Language Models (VLMs) have attracted considerable attention, due to their remarkable performance on a wide range of Vision-Language tasks, including image-text

retrieval, visual entailment and visual grounding [10, 26, 41]. However, existing research has revealed that VLMs are vulnerable to adversarial examples [40]. By making imperceptible modifications to benign images and altering a small portion of the words in benign text, the crafted adversarial examples can easily mislead state-of-the-art VLMs. Importantly, adversarial examples generated against one VLM can still mislead other VLMs due to their transferability, even if these models use different architectures or are used for different tasks [17]. The transferability of adversarial examples makes it feasible to perform more practical black-box attacks, which poses a serious security risk for the deployment of VLMs in safety-critical scenarios.

In this paper, we primarily investigate the transferability of adversarial examples across different VLMs. Different from unimodal models, VLMs can handle both image and text modalities simultaneously as well as image-text pairs exhibit intrinsic alignment and complementarity to each other, making it impractical to employ transferable unimodal attack methods directly. Perturbing image and text independently without considering their interactions may cause the attack to fail due to the two unimodal perturbations conflicting with each other. To consider interactions between attacks of different modalities, existing transferable multimodal attacks collaboratively instead of independently perturb image-text pairs. Moreover, VLMs typically involve a range of non-classification tasks such as cross-modal retrieval, so it is appropriate to conduct attacks against the embedding representations instead of the downstream task labels.

Although cross-modal interactions turn out to be effective in improving the transferability of adversarial examples across VLMs, existing transferable multimodal attack methods have limited consideration for them. Specifically, these methods solely utilize the paired information from another modality as supervision to guide the optimization of the adversarial direction for one modality, which does not adequately model the correspondence between images and their corresponding text. Based on this, we believe that taking more fused information between image and text modalities into account could further improve the transferable attack performance against VLMs. Recently, Stable Diffusion [25] has come to the fore due to accurately and conveniently manipulating the given image with the guidance of a text prompt. Given the cross-attention modules in Stable Diffusion capable of facilitating cross-modal interaction, we believe that leveraging Stable Diffusion to craft adversarial examples could further fuse information between image and text modalities, thereby improving the transferable attack performance against VLMs. Nevertheless, existing transferable diffusion-based attacks mainly focus on unimodal models [1] yet are seldom explored on VLMs.

Based on the analysis mentioned above, we propose an unrestricted attack framework on different Vision-and-Language tasks, named Multimodal Diffusion-based Attack (MDA). Specifically,

MDA initially utilizes the pairwise benign text and image as a guide for generating adversarial text, followed by inversing the benign image into latent space, and applying modifications in latent space influenced by adversarial text, to generate a high transferable adversarial image with satisfactory imperceptibility. To preserve the structure and appearance of images, MDA imposes constraints on the self-attention maps, which have been demonstrated to regulate structure effectively [1]. To preserve the original content structure of adversarial images, MDA imposes constraints on the self-attention maps of Stable Diffusion, which have been demonstrated effective for structure retention [1].

Compared with existing transferable multimodal attack methods, MDA optimizes the image latent of off-the-shelf Stable Diffusion instead of directly manipulating image pixels, resulting in introducing distortions in high-level semantics. MDA not only utilizes the adversarial text to calculate the gradients used for updating adversarial image, but also takes it as the prompt with its high-level semantics to guide the adversarial image generation during the denoising process, which significantly enhances the cross-modal interactions, thus leading high transferability. To evaluate the attack performance of the proposed MDA, we conduct extensive experiments on image-text and text-image retrieval, visual entailment and visual grounding. Experimental results show that our proposed MDA outperforms the current state-of-the-art baselines by a large margin(e.g., in image-text retrieval task, the transferable attack success rate of MDA is more than 30% higher than that of SGA on Flickr30K dataset), showing great performance on adversarial transferability. We also evaluate the cross-task transferability of MDA and other baselines, and the results demonstrate that besides cross-model settings, MDA is also highly transferable in cross-task settings compared with other baselines. Besides listing experimental results, we also exhibit visualizations of the adversarial images and analyze their characteristics on imperceptibility.

The major contributions of this paper could be summarized as follows.

- To the best of our knowledge, we are the first to investigate multimodal unrestricted attack on VLMs via Stable Diffusion and propose a novel attack framework, named Multimodal Diffusion-based Attack(MDA).
- To address the insufficiency of cross-modal interactions in existing multimodal attack methods, MDA strengthens the cross-modal interactions by shifting image latent gradually to adversarial semantic space with the adversarial text guidance, thus enhancing the adversarial transferability.
- We conduct extensive experiments on several multimodal tasks and achieve state-of-the-art transferable attack performance under black-box settings, surpassing other attack methods by a large margin.

## 2 RELATED WORK

### 2.1 Adversarial Attacks

Adversarial attack is first proposed in computer vision highlighting the vulnerability of deep learning models [28]. In the field of attacks on VLMs, Co-attack [40] collectively carries out the attacks on the image modality and the text modality, based on the observation

that bi-modal perturbation is stronger than single-modal perturbation. SGA [17] used multi-scale images and all paired texts to generate adversarial examples, leveraging the richness of data to increase adversarial transferability. Inspired by optimal transport theory [31], OT-Attack [4] uses a more balanced image-text match to conduct attack. SA-Attack [5] enhanced the transferability by utilizing EDA [34] and SIA [33] during the process of adversarial example generation. Although these methods have been explored to enhance the success rate of transfer attacks on VLMs, the adversarial transferability is still limited due to their pixel-based nature. When compared to white-box attacks, there is still a significant drop in attack success rate under black-box settings.

### 2.2 Vision-Language Models

Pre-trained Vision-Language Models (VLMs) represent a class of neural network architectures designed to learn rich cross-modal representations by jointly processing visual and linguistic data. These models aim to capture the inherent interdependencies between visual content, such as images or videos, and textual information, including captions, descriptions, or natural language questions. Most early works on VLMs used pre-trained object detectors to capture local features [14, 30, 32]. Since there has been an increasing interest in Vision Transformers (ViTs) [3] recently, numerous studies based on ViTs have emerged [11, 12, 24], proposing an end-to-end transformation of the input images into patches, thereby enhancing the inference speed. In this work, same as relevant prior works [4, 5, 17, 40], we adopt ALBEF [12], TCL [36] and CLIP [24] as inference models. ALBEF and TCL receive inputs of image and text modalities simultaneously and process these two modalities with their respective encoders to get visual and textual features. Then, the visual and textual features are sent into multimodal fusion module to get a fused multimodal feature. In contrast, CLIP outputs two unimodal features instead of a fused one. According to the categorization in [40], ALBEF and TCL belong to fused models, while CLIP belongs to aligned models.

### 2.3 Vision-Language Tasks

Vision-Language tasks encompass a broad category of problems that involve the integration and interaction of visual and textual information. In this work, we choose image-text retrieval, text-image retrieval, visual grounding and visual entailment as our downstream tasks. Brief introductions about these tasks are listed below:

**Image-Text and Text-Image Retrieval.** In image-text retrieval, an image is sent to an inference model to retrieve the most relevant text, while in text-image retrieval, text is the input and image is the output. For fused models(ALBEF and TCL), feature similarity score is computed using the output of image and text encoder across all image-text pairs. The top-k candidates are then sent into multimodal encoder to calculate the matching score and ranked. For aligned models, since there is no multimodal encoder, the matching score is calculated and ranked based on the output of two unimodal encoders.

**Visual Entailment.** The goal of visual entailment is to predict whether the hypothesis(text) could be reasoned with the information provided by the premise(image), which is proposed in [35].

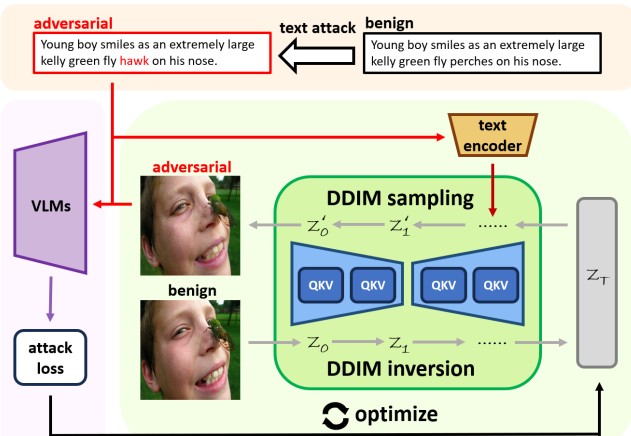

**Figure 1: The framework of MDA. In MDA, adversarial text is generated via text attack at first. After the adversarial text is obtained, Stable Diffusion is then used to generate adversarial image with the guidance of adversarial text as the prompt. Subsequently, the adversarial image-text pair is sent into VLM and the gradients for updating latent $z_T$ is calculated according to the downstream task loss. $z_T$ is then used to generate adversarial image in the next iteration. Update for $z_T$ and adversarial image generation are repeated multiple times. DDIM is leveraged to accelerate the diffusion process.**

Visual entailment could be seen as a classification task that contains three classes: Entailment, Neutral and Contradiction. In ALBEF and TCL, a fully connected layer is used to transform [CLS] token to class probabilities.

**Visual Grounding.** The goal of visual grounding is to identify and locate the objects or regions in the visual data that correspond to the textual description provided. In ALBEF, attention map is used to rank the detected proposals [37].

## 3 METHOD

### 3.1 Problem Formulation

In Vision-Language tasks, we denote the image input as $\mathbf{x}_{img}$ and the text input as $\mathbf{x}_{txt}$. The goal of the transferable multimodal adversarial attack is to make modifications to benign image-text pair yielding adversarial image-text pair $\{\mathbf{x}'_{img}, \mathbf{x}'_{txt}\}$. The optimization problem for generating adversarial image-text pair on surrogate pre-trained VLM $f(\cdot, \cdot)$ is formulated as follows:

$$\max_{(\mathbf{x}'_{img}, \mathbf{x}'_{txt})} \mathcal{L}(\mathbf{x}'_{img}, \mathbf{x}'_{txt}; f), \tag{1}$$

where $\mathcal{L}(\cdot, \cdot)$ denotes the loss function respective of the downstream Vision-Language tasks. The generated adversarial image-text pairs can mislead other target VLMs $g(\cdot, \cdot)$ into $g(\mathbf{x}'_{img}, \mathbf{x}'_{txt}) \neq g(\mathbf{x}_{img}, \mathbf{x}_{txt})$ without knowledge about $f(\cdot, \cdot)$.

Cross-modal interactions can significantly enhance the transferability of multimodal attacks in existing works [17]. However, the simple gradient calculation of the adversarial text during the

image adversarial attack process in existing works [40] renders the under-utilization of cross-modal interactions. Further effort is needed to facilitate deeper cross-modal interactions. We leverage the diffusion model to generate unrestricted adversarial images. Guided by adversarial text during each denoising step, the generated adversarial image with further cross-modal interactions highly boosts adversarial transferability.

### 3.2 Multimodal Diffusion-based Attack (MDA)

In this section, we propose Multimodal Diffusion-based Attack (MDA), a highly transferable multimodal unrestricted attack framework, as displayed in Figure 1. Specifically, we utilize the Text Modality Attack (Section 3.2.1) to generate adversarial text. Then adversarial image is generated in the diffusion process with the guidance of the adversarial text (Section 3.2.2). The algorithm for MDA is presented in Algorithm 1. In the following, the framework will be elaborated in detail.

*3.2.1 Text Modality Attack.* For text attack, our goal is to craft an adversarial text that is most semantically dissimilar from the benign text. Since text space is discrete, unlike attacks on pixel space (e.g., FGSM [28] or PGD [18]), we cannot directly modify tokens in the text based on the gradients obtained from the loss of downstream tasks. Alternatively, we choose the tokens that cause the most significant change in the feature space compared to the benign text. For aligned models like CLIP [24] which is capable of processing text-only input, the aforementioned text attack can be represented by the following equations:

$$\max_{\mathbf{x}'_{txt}} (KL(E_t(\mathbf{x}'_{txt}), E_t(\mathbf{x}_{txt}))), \tag{2}$$

where $E_t(\cdot)$ denotes text embedding and $KL(\cdot, \cdot)$ is Kullback-Leibler divergence, which is adopted to quantify the distance between the features of benign and adversarial text. For fused models like ALBEF [12] and TCL [36], since image and text modalities are designed to be sent into models simultaneously, the text attack is conducted as follows:

$$\max_{\mathbf{x}'_{txt}} (KL(E_m(E_t(\mathbf{x}'_{txt}), E_i((x_i)), E_m(E_t(\mathbf{x}'_{txt}), E_i((x_i))))), \tag{3}$$

where $E_i(\cdot)$ denotes image embedding and $E_m(\cdot, \cdot)$ denotes the multimodal embedding. In this work, we adopt BERT-Attack [13] to conduct adversarial attack on text modality, which has been consistently utilized in prior studies [4, 5, 17, 40] for conducting multimodal attacks.

*3.2.2 Image Modality Attack.* In this work, we employ Stable Diffusion [25] which has been pre-trained on extensive image-text pairs to conduct image modality attack. Stable Diffusion contains multiple cross-attention modules, and could significantly facilitate cross-modal interactions. Specifically, cross-modal interactions happen when the embeddings of the visual and textual features are fused through attention modules during noise prediction, and subsequently boost adversarial transferability.

The procedure for conducting image modality attack is as follows. After adversarial text is generated, it is taken as a prompt and used to guide adversarial image generation in the diffusion process.

Given that adversarial attacks are designed to mislead the target model by adding perturbations to the benign image, they can be considered a special kind of image editing. In the same manner as other existing image editing methods [2, 20, 22], we adopt DDIM Inversion [27] (Please see supplementary materials for details.) to map the benign image into latent space. Let $V_E(\cdot)$ be the VAE encoder of Stable Diffusion, the inversion process is conducted by the following equation:

$$\mathbf{z}_0 = V_E(\mathbf{x}_{img}), \quad \mathbf{z}_T = \text{In}(\mathbf{z}_{T-1}) = \underbrace{\text{In} \circ \cdots \circ \text{In}}_{T}(\mathbf{z}_0), \quad (4)$$

and $\text{In}(\cdot)$ in Equation 4 is denoted as follows:

$$\text{In}(\mathbf{z}_t) = \sqrt{\frac{\bar{\alpha}_{t+1}}{\bar{\alpha}_t}}\mathbf{z}_t + \sqrt{\mathbf{z}_{t+1}}\left(\sqrt{\frac{1}{\bar{\alpha}_{t+1}-1}} - \sqrt{\frac{1}{\bar{\alpha}_t-1}}\right)\epsilon_\theta(\mathbf{z}_t, t, \varnothing). \quad (5)$$

In the preceding equation, $\mathbf{z}_t$ denotes the latent code, $\bar{\alpha}_t$ denotes the noise scaling factor and $\epsilon_\theta(\mathbf{z}_t, t, \varnothing)$ denotes the predicted noise added in $\mathbf{z}_t$ with null text $\varnothing = $ "", respectively. $t$ is a certain timestep and $T$ is the total number of timesteps. According to the study in [20], an slight error is accumulated in each step of DDIM inversion, but the accumulated error is negligible for unconditional diffusion models, thus null text is chosen in the DDIM inversion of MDA instead of clean text.

After obtaining the reversed latent $\mathbf{z}_T$, the denoising process is conducted to get the generated image $\mathbf{x}'_{img}$ with the guidance of adversarial text:

$$\mathbf{z}'_T = \mathbf{z}_T, \quad \mathbf{z}'_0 = \text{De}(\mathbf{z}'_1) = \underbrace{\text{De} \circ \cdots \circ \text{De}}_{T}(\mathbf{z}'_T), \quad \mathbf{x}'_{img} = V_D(\mathbf{z}'_0). \quad (6)$$

Here $V_D(\cdot)$ is the decoder of VAE used to decode the image from latent space to pixel space, and $\text{De}(\cdot)$ in Equation 6 is denoted as follows:

$$\text{De}(\mathbf{z}'_t) = \sqrt{\frac{\bar{\alpha}_{t-1}}{\bar{\alpha}_t}}\mathbf{z}'_t + \sqrt{\mathbf{z}'_{t-1}}\left(\sqrt{\frac{1}{\bar{\alpha}_{t-1}-1}} - \sqrt{\frac{1}{\bar{\alpha}_t-1}}\right)\tilde{\epsilon}_\theta(\mathbf{z}'_t, t, \mathbf{x}'_{txt}, \varnothing), \quad (7)$$

The classifier-free guidance(CFG) technique[8] is proposed as a way to control the amount of weight the model gives to the conditioning information with guidance scale parameter $\omega$. CFG could be regarded as an approach of adjusting the degree of cross-modal interactions therefore is adopted during the denoising process of our attack. CFG is expressed in Equation 7 as follows:

$$\tilde{\epsilon}_\theta(\mathbf{z}'_t, t, C, \varnothing) = \omega \cdot \epsilon_\theta(\mathbf{z}'_t, t, C) + (1-\omega) \cdot \epsilon_\theta(\mathbf{z}'_t, t, \varnothing). \quad (8)$$

For better image reconstruction quality, we adopt the method proposed in [20] to optimize the null text embedding before generating adversarial images. After $\mathbf{x}'_{img}$ is generated, gradients could be calculated with the loss function $\mathcal{J}(\cdot, \cdot)$ respective of downstream Vision-Language tasks to update $\mathbf{z}_T$. The subsequent equation represents the optimization goal:

$$\mathcal{L}_{attack} = -\mathcal{J}(\mathbf{x}'_{img}, \mathbf{x}'_{txt}). \quad (9)$$

Since our method is generative, it is necessary to prevent excessive alterations between the benign and adversarial images; otherwise, the original semantic structure will be destroyed and the adversarial image will look far different from the benign one, in which case the adversarial attack generates an irrelevant image and makes no sense. To preserve the original semantic structure, we adopt the self-attention constraint introduced in [1], in which self-attention maps are utilized to restrict the degree of modifications between clean and adversarial latents. The corresponding structure preserving loss is:

$$\mathcal{L}_{structure} = \|sa_{clean} - sa_{adv}\|_2^2, \quad (10)$$

where $sa_{clean}$ and $sa_{adv}$ represent the self-attention maps for the benign and adversarial latents, respectively. Along with equation 9, the total optimization objective is:

$$\min_{\mathbf{z}'_T} \mathcal{L}_{total} = \mu\mathcal{L}_{attack} + \mathcal{L}_{structure}, \quad (11)$$

where $\mu$ is used to control the relative weights of attack loss and structure preserving loss. $\mathbf{z}'_T$ is updated for multiple times before the adversarial example is determined. In Equation 11 we leave out the notation denoting iterations for simplicity.

According to [1, 19], the diffusion models exhibit a propensity to concentrate on coarse semantic information in the early steps and fine-grained information in the later steps. Besides, a larger number of total timesteps could significantly impact the image generation, enhancing the attack strength yet reducing the image quality. Thus, we follow [1] and apply limited DDIM inversion steps at the back of denoising process to preserve high-level semantics.

---

**Algorithm 1** Multimodal Diffusion-bansed Attack(MDA)

1: **Input:** benign example $(\mathbf{x}_{img}, \mathbf{x}_{txt})$, Vision-Language model $f$, total timestep of DDIM $T$, attack iterations $I$, VAE encoder $V_E(\cdot)$, VAE decoder $V_D(\cdot)$, inverse process $In(\cdot)$, denoise process $De(\cdot)$
2: **Output:** Adversarial example $(\mathbf{x}'_{img}, \mathbf{x}'_{txt})$
3: $\mathbf{z}_0 = V_E(\mathbf{x}_{img})$
4: **for** $t = 1, \ldots, T$ **do**
5: $\quad \mathbf{z}_t = In(\mathbf{z}_{t-1})$
6: **end for**
7: $\mathbf{z}'_T = \mathbf{z}_T$
8: **for** $t = T, \ldots, 1$ **do**
9: $\quad$ optimize null text embedding $\varnothing_t$
10: **end for**
11: $\mathbf{x}'_{txt} = $ BERT-Attack$(\mathbf{x}_{txt})$
12: **for** $i = 1, \ldots, I$ **do**
13: $\quad$ **for** $t = T, \ldots, 1$ **do**
14: $\quad\quad \mathbf{z}'_{t-1} = De(\mathbf{z}'_t)$
15: $\quad$ **end for**
16: $\quad \mathbf{x}'_{img} = V_D(\mathbf{z}'_0)$
17: $\quad$ calculate $\mathcal{L}_{attack}$ using Eq. 9
18: $\quad$ calculate $\mathcal{L}_{structure}$ using Eq. 10
19: $\quad$ update $\mathbf{z}'_T$ over $\mathcal{L}_{total}$ with AdamW optimizer
20: **end for**
21: $\mathbf{x}'_{img} = V_D(\mathbf{z}'_0)$
22: **return** $(\mathbf{x}'_{img}, \mathbf{x}'_{txt})$

# 4 EXPERIMENTS

## 4.1 Experiment Settings

**Datasets.** For image-text and text-image retrieval task, we consider two commonly used datasets: Flickr30K [23] and MSCOCO [15] dataset. Flickr30K dataset contains 31783 images, and each image corresponds to five captions. MSCOCO dataset contains 123,287 images, and each image corresponds to approximately five captions. For visual entailment task, we use SNLI-VE [35] dataset. Images in SNLI-VE dataset are the same as those in Flickr30K dataset and correspond to varied number of captions. For visual grounding task, we use RefCOCO+ [38] dataset, which contains 49856 object entities, 141,564 captions and 19,992 images.

**Vision-Language Models.** We follow previous studies [4, 5, 17, 40] and evaluate three popular VLMs, including ALBEF [12], TCL [36] and CLIP [24]. ALBEF is a Vision-Language model constructed on a transformer-based architecture, composed of an image encoder, a text encoder and a multimodal encoder. TCL shares a common architecture with ALBEF, but is trained with different strategies including cross-modal alignment, intra-modal contrastive, and local mutual information maximization. CLIP has two implementations, one with ResNet-101 [6] and the other ViT/B-16 [3] as the image encoder. These two implementations are denoted $CLIP_{CNN}$ and $CLIP_{ViT}$, respectively.

**Implementation Details.** We adopt BERT-Attack [13] to generate text adversarial examples with a text perturbation bound $\epsilon_t$ of 1 token and word list length $W_l = 10$. Images are resized to a resolution of 224x224 pixels before fed into ALBEF or TCL, and 384x384 pixels before fed into CLIP. DDIM [27] with 20 steps is implemented as the sampler of Stable Diffusion [25], and DDIM inversion steps is set to 5. The guidance scale $\omega$ of Stable Diffusion is set to 2.5. The size of images generated by Stable Diffusion is 224x224. The number of iterations of updating latent is 30 and AdamW [16] with learning rate of 0.01 is adopted as the optimizer. The relative weight control factor $\mu$ is 0.5. The image perturbation bound $\epsilon_i$ of all pixel-based attacks used in the following experiments is set to 2/255. All experiments are performed on a single RTX 3090 GPU.

**Evaluation Metrics.** Attack Success Rate (ASR) is employed to measure the adversarial robustness of the target models. The calculation method of ASR varies in different tasks. A higher ASR indicates better attack performance. We adopt Frechet Inception Distance (FID) [7] to measure the natrualness of the crafted adversarial images. FID is calculated between the adversarial images and the benign images from the corresponding validation set. A lower FID indicates better image quality. We also adopt NIMA-AVA [29] trained on AVA dataset [21] for image quality assessment. A higher NIMA-AVA indicates better image quality.

## 4.2 Attack Performance on Image-Text and Text-Image Retrieval

In order to test the attack performance of MDA, we first conduct experiments on image-text and text-image retrieval tasks on Flickr30K dataset. Besides MDA, we also adopt PGD [18], BERT-Attack [13], Sep-Attack [17], Co-Attack [40] and Set-level Guidance Attack (SGA) [17] as the compared attacks. ASR based on R@K (R stands

for recall) is used as the metric and the value of K is set to 1, 5, 10 in the experiments. Results on ASR are listed in Table 1 and Table 2.

Tables 1 and Table 2 demonstrate the significant superiority of our method over all other attacks in both image-text and text-image retrieval tasks under black-box setting. For instance, in image-text retrieval task, the ASR on TCL with adversarial examples generated on ALBEF using our method surpass those of SGA by more than 30% (76.29% vs. 45.42% for R@1, 59.20% vs. 24.93% for R@5, and 51.10% vs. 16.48% for R@10, respectively). Adversarial examples generated on ALBEF (aligned VLM) also achieve high transfer attack performance on $CLIP_{ViT}$ and $CLIP_{CNN}$ (fused VLMs), which indicates that our MDA could provide strong cross-architecture adversarial transferability. The exceptional performance of our method can be attributed to several key factors. Firstly, in addition to using adversarial text to compute the gradients necessary for updating the adversarial image, our MDA leverages adversarial text as a prompt to guide the generation of adversarial images. This approach allows us to maximize the utilization of adversarial text compared to other methods. Furthermore, Stable Diffusion, employed in MDA, incorporates transformer modules consisting of multiple cross-attention blocks, facilitating extra cross-modal interactions. By enhancing these interactions and maximizing the use of adversarial text, our method achieves higher adversarial transferability. In addition to enhancing cross-modal interactions, the design of MDA incorporates several other factors that contribute to improving adversarial transferability. By perturbing the latent space instead of pixel space, MDA introduces distortions in high-level semantics, which has been shown to enhance adversarial transferability according to recent studies [9, 39]. Furthermore, MDA includes an additional denoising process and leverages transformations provided by Stable Diffusion that are unrelated to gradients calculated using the loss of downstream task. This feature helps mitigate overfitting of the surrogate model. However, it is important to note that the performance of MDA is slightly lower in ASR compared to SGA under white-box settings. This discrepancy is attributed to the fact that generative diffusion-based attacks rely less on the surrogate model, resulting in poorer white-box attack performance.

Figure 2 further shows several attack examples on the image-text retrieval results. From the results, it is evident that by manipulating subtle content, our MDA method effectively deceives the retrieval model, resulting in incorrect top-1 retrieved text that does not match the query image. It is worthwhile mentioning that the alternations in the generated adversarial images are hardly noticeable by humans.

We then measure the image quality of the adversarial examples generated by these attacks using FID and NIMA-AVA, and the results are listed in Table 3. It is important to note that the magnitude of modifications in pixel-based attacks and unrestricted attacks is typically evaluated using different ways (perturbation bound for pixel-based attacks and image quality for unrestricted attacks). Hence it is hard to choose a fair metric to evaluate pixel-based attacks along with unrestricted attacks. From the results, we have the following observations. Basically, for FID value, our MDA is slightly higher than other baselines while for MIMA-AVA metric, our MDA is slightly lower than the other baselines. Considering the fact that MDA surpasses other attacks in ASR by a large margin, the image quality results of adversarial examples generated by MDA

**Table 1: ASR on image-text retrieval task on Flickr30K dataset. * indicates the performance under white-box attack. The best results are highlighted in bold.**

| Surrogate Model | Attack | ALBEF | | | TCL | | | CLIP$_{ViT}$ | | | CLIP$_{CNN}$ | | |
|---|---|---|---|---|---|---|---|---|---|---|---|---|---|
| | | R@1 | R@5 | R@10 | R@1 | R@5 | R@10 | R@1 | R@5 | R@10 | R@1 | R@5 | R@10 |
| ALBEF | PGD | 52.45* | 36.57* | 30.00* | 3.06 | 0.40 | 0.10 | 8.96 | 1.66 | 0.41 | 10.34 | 2.96 | 1.85 |
| | BERT-Attack | 11.57* | 1.80* | 1.10* | 12.64 | 2.51 | 0.90 | 29.33 | 11.63 | 6.30 | 32.69 | 15.43 | 8.65 |
| | Sep-Attack | 65.69* | 47.60* | 42.10* | 17.60 | 3.72 | 1.90 | 31.17 | 12.05 | 7.01 | 32.82 | 15.86 | 9.06 |
| | Co-Attack | 77.16* | 64.60* | 58.37* | 15.21 | 4.19 | 1.47 | 23.60 | 7.82 | 3.93 | 25.12 | 8.42 | 5.39 |
| | SGA | **97.24*** | **94.09*** | **92.30*** | 45.42 | 24.93 | 16.48 | 33.38 | 13.50 | 9.04 | 34.93 | 17.07 | 10.45 |
| | MDA | 93.53* | 87.47* | 83.2* | **76.29** | **59.20** | **51.10** | **70.18** | **46.83** | **38.41** | **73.31** | **50.21** | **39.55** |
| TCL | PGD | 6.15 | 1.30 | 0.70 | 77.87* | 65.13* | 58.72* | 7.48 | 1.45 | 0.81 | 10.34 | 2.75 | 1.54 |
| | BERT-Attack | 11.89 | 2.20 | 0.70 | 14.54* | 2.31* | 0.60* | 29.69 | 12.77 | 7.62 | 33.46 | 14.38 | 9.37 |
| | Sep-Attack | 20.13 | 4.91 | 2.70 | 84.72* | 73.07* | 65.43* | 31.29 | 12.98 | 7.72 | 33.33 | 14.27 | 9.89 |
| | Co-Attack | 23.15 | 6.98 | 3.63 | 77.94* | 64.26* | 56.18* | 27.85 | 9.80 | 5.22 | 30.74 | 12.09 | 7.28 |
| | SGA | 48.91 | 30.86 | 23.10 | **98.37*** | **96.53*** | **94.98*** | 33.87 | 15.21 | 9.46 | 37.74 | 17.86 | 11.74 |
| | MDA | **81.86** | **70.64** | **65.60** | 91.78* | 87.94* | 85.47* | **74.72** | **55.04** | **44.51** | **75.73** | **57.29** | **49.33** |

**Table 2: ASR on text-image retrieval task on Flickr30K dataset. * indicates the performance under white-box attack. The best results are highlighted in bold.**

| Surrogate Model | Attack | ALBEF | | | TCL | | | CLIP$_{ViT}$ | | | CLIP$_{CNN}$ | | |
|---|---|---|---|---|---|---|---|---|---|---|---|---|---|
| | | R@1 | R@5 | R@10 | R@1 | R@5 | R@10 | R@1 | R@5 | R@10 | R@1 | R@5 | R@10 |
| ALBEF | PGD | 58.65* | 44.85* | 38.98* | 6.79 | 2.21 | 1.20 | 13.21 | 5.19 | 3.05 | 14.65 | 5.60 | 3.39 |
| | BERT-Attack | 27.46* | 14.48* | 10.98* | 28.07 | 14.39 | 10.26 | 43.17 | 26.37 | 19.91 | 46.11 | 28.43 | 22.14 |
| | Sep-Attack | 73.95* | 59.50* | 53.70* | 32.95 | 17.10 | 11.90 | 45.23 | 25.93 | 19.95 | 45.49 | 28.43 | 22.32 |
| | Co-Attack | 83.86* | 74.63* | 70.13* | 29.49 | 14.97 | 10.55 | 36.48 | 21.09 | 15.76 | 38.89 | 22.38 | 17.49 |
| | SGA | **97.28*** | **94.27*** | **92.58*** | 55.25 | 36.01 | 27.25 | 44.16 | 27.35 | 20.84 | 46.57 | 29.16 | 22.68 |
| | MDA | 93.94* | 88.58* | 85.18* | **79.90** | **64.52** | **57.05** | **73.78** | **57.32** | **48.93** | **75.13** | **58.90** | **49.90** |
| TCL | PGD | 10.78 | 3.36 | 1.70 | 79.48* | 66.26* | 60.36* | 13.72 | 5.37 | 3.01 | 15.33 | 5.77 | 3.28 |
| | BERT-Attack | 26.82 | 14.09 | 10.80 | 29.17* | 15.03* | 10.91* | 44.49 | 27.47 | 21.00 | 46.07 | 29.28 | 22.59 |
| | Sep-Attack | 36.48 | 19.48 | 14.82 | 86.07* | 74.67* | 68.83* | 44.65 | 26.82 | 20.37 | 45.80 | 29.18 | 23.02 |
| | Co-Attack | 40.04 | 22.66 | 17.23 | 85.59* | 74.19* | 68.25* | 41.19 | 25.22 | 19.01 | 44.11 | 26.67 | 20.66 |
| | SGA | 60.34 | 42.47 | 34.59 | **98.81*** | **97.19*** | **95.86*** | 44.88 | 28.79 | 21.95 | 48.30 | 29.70 | 23.68 |
| | MDA | **85.62** | **74.90** | **69.23** | 93.69* | 89.35* | 86.98* | **78.03** | **62.98** | **54.30** | **79.21** | **64.34** | **55.88** |

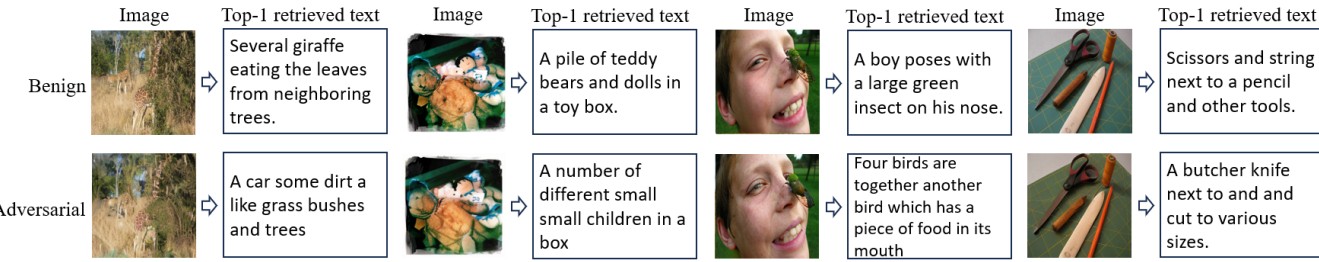

**Figure 2: Attack examples of our method in image-text retrieval task.**

is indeed acceptable. We also conduct experiments on MSCOCO dataset, and the results are shown in supplementary materials. Basically, similar observations can be obtained from the results.

## 4.3 Attack Performance on Visual Entailment

In addition to the cross-modal retrieval task mentioned above, we also evaluate the proposed method on visual entailment task. The evaluation is conducted by generating adversarial examples on

**Table 3: Evaluation on the quality of adversarial examples generated by different attack methods on Flickr30K dataset.**

**(a) Comparison on FID. A lower FID indicates better image quality.**

| Attack method
Surrogate model | PGD | Co-Attack | SGA | MDA |
|---|---|---|---|---|
| ALBEF | 58.75 | 58.79 | 59.71 | 67.78 |
| TCL | 58.65 | 58.70 | 59.68 | 66.89 |

**(b) Comparison on NIMA-AVA. A higher NIMA-AVA indicates better image quality.**

| Attack method
Surrogate model | PGD | Co-Attack | SGA | MDA |
|---|---|---|---|---|
| ALBEF | 5.90 | 5.90 | 5.89 | 5.57 |
| TCL | 5.90 | 5.90 | 5.87 | 5.59 |

ALBEF/TCL and then transfer to attack TCL/ALBEF. ASR on classification accuracy is adopted as the metric to evaluate attack performance. For each image, we randomly choose one corresponding text to construct image-text pairs. The attack results are presented in Figure 3. From the results, we can observe that our method surpasses all other attack methods in both ALBEF-to-TCL and TCL-to-ALBEF settings, achieving the best attack performance. The results indicate that our method achieves high transferability across different downstream Vision-Language tasks, which is due to the additional cross-modal interactions provided in our method.

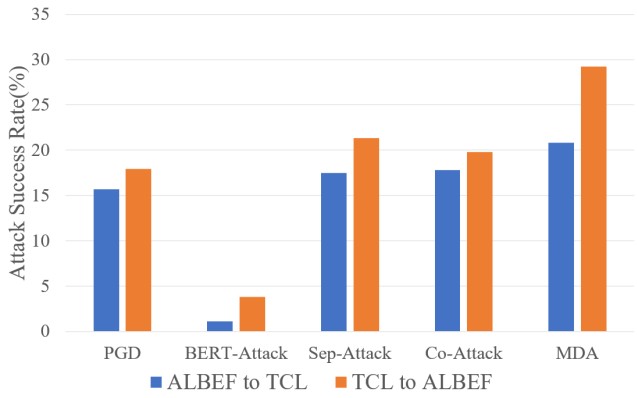

**Figure 3: ASR on visual entailment task on SNLI-VE dataset. Adversarial examples are generated on ALBEF (TCL) and transfered to TCL (ALBEF).**

### 4.4 Cross-Task Transferability

We also conducted experiments to assess cross-task transferability. Following the previous experimental setup of [17], we evaluated the attack performance on the visual grounding task using adversarial examples generated from the image-text retrieval task. The results are summarized in Table 4. From the results, our method

continues to outperform the state-of-the-art attack SGA by a significant margin (55.02% vs. 61.19% on test-A and 40.19% vs. 43.71% on test-B). The results indicate that MDA achieves high adversarial transferability in both cross-model and cross-task settings.

**Table 4: Accuracy of visual grounding on RefCOCO+ dataset. The adversarial examples used for evaluation are generated on image-text retrieval task, thus capable of measuring cross-task transferability. The target model and surrogate model are both ALBEF. Baseline represents the accuracy on clean data. Lower values indicate better cross-task transferability.**

| Test set
Attack method | test-A | test-B |
|---|---|---|
| Baseline | 65.89 | 46.25 |
| Co-Attack | 61.80 | 43.81 |
| SGA | 61.19 | 43.71 |
| MDA | **55.02** | **40.19** |

### 4.5 Ablation Studies

We then investigate the effect of different parameter settings on attack success rates using Flickr30K dataset, here ALBEF is utilized as the surrogate model. We first conduct experiments with different relative weight control factor $\mu$ varying from 0.1 to 0.5 and a fixed guidance scale $\omega$ at 2.5. The ASR results based on TR@1(image-text R@1) and IR@1(text-image R@1) are listed in Table 5. Increasing $\mu$ indicates paying more attention on the attack loss rather than the retention of semantic structure, which notably affects both white-box and black-box results. For instance, when $\mu$ is set to 0.1, the white-box ASR is 67.26% on TR@1 and 73.71% on IR@1. Conversely, when $\mu$ is increased 0.5, the white-box ASRs on TR@1 and IR@1 are significantly increased to 93.53% and 93.94%, respectively. Similarly, When taking TCL as the target model, increasing $\mu$ from 0.1 to 0.5 results in a notable increase in ASR (from 47.42% to 76.29% on TR@1 and 59.36% to 79.90% on IR@1). It's worth noting that there is a slight increase in FID of adversarial examples as $\mu$ becomes larger, which is an expected outcome of enhancing the attack strength.

In addition to the relative weight control factor $\mu$, altering the guidance scale $\omega$ also impacts the attack performance, as shown in Table 6. A higher value assigned to the guidance scale $\omega$ results in more robust guidance provided by the adversarial text during the denoising process. The table illustrates a significant increase in both white-box and black-box ASR, along with a slight rise in FID, with a larger guidance scale. These findings suggest a trade-off between attack effectiveness and image fidelity.

### 4.6 Visualization

We further visualize several benign images and their corresponding adversarial images generated using our method in Figure 4. It is evident that the modifications in the generated adversarial image are subtle compared to the original clean image. Basically, the alterations induced by our MDA method primarily involve modifications to the texture, edges, and color of the objects in the image. These changes usually happen in a very small area in the image, hence

**Table 5: Effects of different relative weight control factor $\mu$. * indicates white-box attack.**

| $\mu$ | FID↓ | ALBEF* | | TCL | | CLIP$_{\text{ViT}}$ | | CLIP$_{\text{CNN}}$ | |
|---|---|---|---|---|---|---|---|---|---|
| | | TR@1* | IR@1* | TR@1 | IR@1 | TR@1 | IR@1 | TR@1 | IR@1 |
| 0.1 | 64.34 | 67.26 | 73.71 | 47.42 | 59.36 | 50.06 | 60.50 | 54.66 | 62.71 |
| 0.2 | 65.47 | 78.00 | 83.65 | 59.85 | 67.43 | 55.83 | 66.37 | 62.96 | 68.89 |
| 0.3 | 66.67 | 84.98 | 88.31 | 67.12 | 73.88 | 62.33 | 69.59 | 66.28 | 71.97 |
| 0.4 | 67.22 | 87.38 | 90.11 | 71.02 | 75.48 | 64.17 | 71.04 | 68.20 | 72.73 |
| 0.5 | 67.78 | 93.53 | 93.94 | 76.29 | 79.90 | 70.18 | 73.78 | 73.31 | 75.16 |

**Table 6: Effects of different guidance scale $\omega$. * indicates white-box attack.**

| $\omega$ | FID↓ | ALBEF* | | TCL | | CLIP$_{\text{ViT}}$ | | CLIP$_{\text{CNN}}$ | |
|---|---|---|---|---|---|---|---|---|---|
| | | TR@1* | IR@1* | TR@1 | IR@1 | TR@1 | IR@1 | TR@1 | IR@1 |
| 1 | 65.93 | 88.63 | 91.35 | 65.23 | 70.88 | 59.64 | 65.27 | 59.64 | 65.21 |
| 1.5 | 65.86 | 88.74 | 92.68 | 68.60 | 75.60 | 62.32 | 68.23 | 62.32 | 68.61 |
| 2 | 66.49 | 89.78 | 92.33 | 73.55 | 78.10 | 67.94 | 71.36 | 67.94 | 73.31 |
| 2.5 | 67.78 | 93.53 | 93.94 | 76.29 | 79.90 | 73.31 | 73.78 | 73.31 | 75.13 |

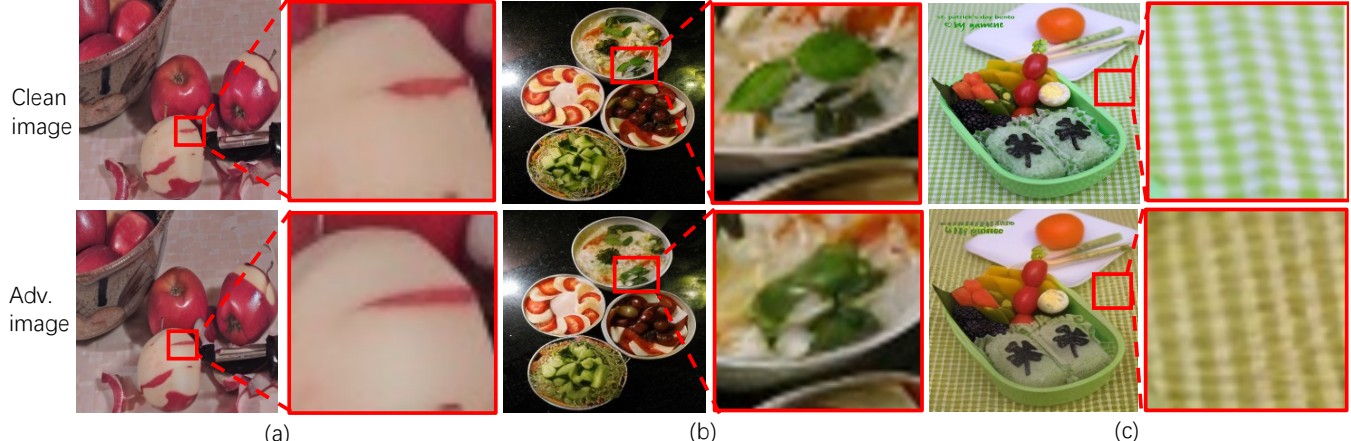

Clean image

Adv. image

(a)                    (b)                    (c)

**Figure 4: Visualization of clean images and their corresponding adversarial images generated using our MDA. Image details are enlarged.**

are hardly perceptible by human eyes. For example, in Figure 4(a) and (b), our MDA generates the adversarial image by changing the shape of the "apple peel" and "mint leaf". In contrast, in Figure 4(c), the adversarial image is generated by altering its background color. Nevertheless, all these changes are subtle and difficult to notice.

## 5  CONCLUSION

In this paper, we explore the potential of Stable Diffusion to conduct transfer-based attack under multimodal settings. We propose MDA, a highly transferable multimodal attack against VLMs, which leverages Stable Diffusion to generate adversarial image with the guidance of adversarial text. MDA uses Stable Diffusion, which contains multiple cross-attention modules, to enable extra cross-modal interactions by taking adversarial text as the prompt to guide adversarial image generation to boost adversarial transferability. Besides, MDA introduces perturbations in the latent space rather than pixel space to manipulate high-level sematics, which also improves adversarial transferability. We conduct extensive experiments on several downstream tasks including cross-modal retrieval, visual entailment and visual grounding. Experimental results show that our proposed MDA surpassing other attacks in black-box settings by a large margin, which demonstrates MDA significantly strengthenes the adversarial transferability. We also exhibit the adversarial examples generated by MDA and analyze their visual characteristics. We hope this work could promote further investigations on the adversarial transferability on VLMs.

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
