# OpenReview forum: "Highly Transferable Diffusion-based Unrestricted Adversarial Attack on Pre-trained Vision-Language Models"
_acmmm.org/ACMMM/2024/Conference — MM2024 Poster_

### Official Review · Reviewer_XgAE · 2024-05-10

**Rating:** 3
**Confidence:** 3

**Summary:**

This paper proposes an MDA that leverages the text prompts to generate the adversarial sample to attack the VLMs.
It first uses the Bert-attack method to generate the adversarial text prompt, then uses the null-text inversion method to edit the images to improve the attack success rate. In the end, adversarial perturbation is introduced to the generated images following the previous method to improve the attack success rate further. The overall method is sound, and using the adversarial text prompt to edit images is interesting. The experimental results show that MDA significantly improves compared to the baselines.
However, it seems that this paper is an incremental work since all the methods used are from the pervious method. Meanwhile, it lacks the SOTA attack methods and the advanced VLMS, which further weakens the contribution of this paper.

**Strengths:**

1. The overall method is sound. This paper first generates an adversarial text prompt and uses it to edit the clean image. The null-text inversion method keeps the overall content unchangeable and only edits a small part. Then, based on the generated image, this paper further introduces the adversarial perturbation based on the previous methods.

2. The proposed MDA makes a significant improvement compared to the baselines.

**Limitations:**

1. The experiments lack SOTA VLM-based attack methods such as Zhao et al. [1]. Meanwhile, verifying the proposed method on advanced VLMs such as BLIP, BILP2, and MiniGPT may be better. Therefore, the overall baselines, including attack methods and the VLMs used in this paper, are insufficient.

[1] On Evaluating Adversarial Robustness of Large Vision-Language Models. Zhao, Yunqing and Pang, Tianyu and Du, Chao and Yang, Xiao and Li, Chongxuan and Cheung, Ngai-Man and Lin, Min, NeurIPS 2023.

2. Typo problems. 1) Eq. 7 is wrong compared to the original paper on de inversion. It should be $z_{t+1} = \sqrt{\frac{\alpha_{t+1}}{\alpha_{t}}}z_{t} + (\sqrt{\frac{1}{\alpha_{t+1}}-1} - \sqrt{\frac{1}{\alpha_{t}}-1})\epsilon_{\theta}$.  2) What is $x_{i}$?  3) Fig. 3 should be improved. At least, the format in the figure should keep the same.

3. The overall novelty seems limited. This paper first uses the Bert-attack method to generate the adversarial prompt and then uses this prompt to generate the adversarial sample. The self-attention control and the attack loss are all from the previous method. The editing process that uses a text prompt to edit prompt is the same as null-text inversion [2]. Therefore, it seems that the overall work is incremental. What is this paper's main contribution?

[2] Null-text Inversion for Editing Real Images using Guided Diffusion Models, Ron Mokady, Amir Hertz, Kfir Aberman, Yael Pritch, Daniel Cohen-Or. CVPR 2023.

4. The author said they set attack bound $\epsilon_{i} = 2/255$ mentioned in Sec. 4. However, the overall method does not need this bound, which confuses. If attack bound is used in this paper, it should make an additional ablation study.

Due to the above weaknesses, we rate it as a borderline reject.

**Suitability:**

3

---

### Official Review · Reviewer_nUmw · 2024-05-23

**Rating:** 3
**Confidence:** 4

**Summary:**

This paper explores the potential of Stable Diffusion to conduct transfer-based attack under multimodal settings.   The authors propose a highly transferable multimodal attack against VLMs, which leverages Stable Diffusion to generate adversarial image with the guidance of adversarial text.

**Strengths:**

(1) The proposed MDA method leverages cross-modal interactions and manipulation of high-level semantics to improve adversarial transferability.
(2) MDA achieves a good balance between adversarial transferability and image quality, as evidenced by the FID and NIMA-AVA evaluation results.

**Limitations:**

1.  This paper focus on Multimodal Unrestricted Adversarial Attack, why there is no the related works such as [1] as a typical Unrestricted Adversarial Attack in your related works section?  If you can explain it, I would change my score.

2.  The attack performance of PGD attack and BERT attack in Figure 3 is very different, and the method proposed in this paper also has a big difference in the performance of the two. What is the reason? It is suggested to add a statement to analyze the phenomenon.

[1] Chen Z, Li B, Wu S, et al. Content-based unrestricted adversarial attack[J]. Advances in Neural Information Processing Systems, 2024, 36.

**Suitability:**

3

---

### Official Review · Reviewer_bsf9 · 2024-05-26

**Rating:** 5
**Confidence:** 3

**Summary:**

This paper presents Multimodal Diffusion-based Attack (MDA), a novel framework for highly transferable adversarial attacks on Vision-Language Models (VLMs). Utilizing Stable Diffusion, MDA introduces adversarial perturbations in the latent space guided by adversarial text, enhancing cross-modal interactions and transferability. It first generates adversarial text, which then guides image generation during the diffusion process. MDA outperforms existing methods in various tasks like image-text retrieval and visual entailment under black-box settings. Its contributions include pioneering multimodal unrestricted attacks via Stable Diffusion and significantly improving adversarial transferability across tasks and models​​.

**Strengths:**

1. MDA enhances adversarial transferability across VLMs by optimizing perturbations in the latent space using Stable Diffusion​​.
2. MDA leverages adversarial text as a prompt, enriching cross-modal interactions and improving transferability of adversarial examples​​.
3. MDA outperforms existing methods in various tasks, demonstrating superior performance under black and white-box settings​​.
4. This paper is well-written and easy to follow.

**Limitations:**

1. Diffusion models have been extensively studied for Adversarial Example Synthesis. What are the differences, necessities, and challenges of adversarial example synthesis in Pre-trained Vision-Language Models?
2. The technical innovation of the article may be somewhat lacking. Could you provide more experimental findings and explanations of technical innovations?

**Suitability:**

3

---

### Official Review · Reviewer_iHRw · 2024-06-06

**Rating:** 4
**Confidence:** 3

**Summary:**

This paper presents a novel adversarial attack framework named Multimodal Diffusion-based Attack (MDA) targeting pre-trained Vision-Language Models (VLMs). Utilizing Stable Diffusion, MDA generates adversarial text to guide the creation of adversarial images by leveraging cross-modal interactions. This approach enhances the transferability of attacks across different VLMs and tasks. Experimental results demonstrate that MDA significantly outperforms existing methods in black-box settings, achieving high attack success rates and maintaining image quality. The framework introduces perturbations in the latent space rather than the pixel space, further improving adversarial transferability.

**Strengths:**

This paper presents a novel adversarial attack framework, Multimodal Diffusion-based Attack (MDA), with several key strengths. Firstly, MDA enhances adversarial transferability across Vision-Language Models (VLMs) by leveraging Stable Diffusion for improved cross-modal interactions, using adversarial text to guide image generation. This approach significantly outperforms existing methods in black-box settings. Secondly, MDA introduces perturbations in the latent space rather than the pixel space, effectively manipulating high-level semantics for better transferability. The framework demonstrates high success rates across multiple tasks, including image-text retrieval, text-image retrieval, visual entailment, and visual grounding. Despite its high effectiveness, MDA maintains the quality of adversarial images, making the changes subtle and hard to detect. Additionally, the paper includes comprehensive experiments and detailed visualizations, underscoring the robustness and practical applicability of MDA. These strengths collectively position MDA as a significant advancement in the field of adversarial attacks on VLMs.

**Limitations:**

This paper has the following limitations:

1. While MDA excels in black-box settings, its performance in white-box settings is slightly lower compared to some existing methods. This indicates potential overfitting issues when the surrogate model is known.

2. The method involves multiple iterations of latent space perturbations and diffusion processes, which can be computationally intensive and time-consuming, requiring significant computational resources.

3. The paper primarily focuses on a few specific Vision-Language tasks and datasets. The generalizability of the results to other tasks or more diverse datasets is not thoroughly explored.

4. The paper does not extensively discuss potential defense mechanisms or robustness improvements that could counteract the proposed attacks, leaving an area of vulnerability unaddressed.

**Suitability:**

3

---

### Meta-Review · Area_Chair_JrrP · 2024-07-01

**Recommendation:** Accept (Poster)
**Confidence:** 5

**Metareview:**

This paper proposed a novel adversarial attack framework named Multimodal Diffusion-based Attack targeting pre-trained Vision-Language Models. The paper received 1 weak accept, 2 borderline accept, and 1 borderline reject. AC looked at the reviews. The main concerns from the reviewer who gave borderline reject are one baseline method and another explanation. AC looked at the rebuttal and the authors have explained the reasons. Therefore, AC voted for the acceptance of this paper.